# Building PhilKG: An LLM-Powered Knowledge Graph from the Stanford Encyclopedia of Philosophy

## Abstract

Philosophical inquiry unfolds as a network of ideas, debates, and thinkers. We present the Philosophy Knowledge Graph, a structured map derived from the complete Stanford Encyclopedia of Philosophy that converts narrative prose into entities and relations suitable for analysis. The construction process is semi automatic: large language models extract people, concepts, and claims from encyclopedia text, and a stronger model reviews selected outputs to confirm support in context. The resulting resource includes over one hundred forty thousand nodes and more than one hundred thousand links, enabling querying, exploration, and comparative study. We illustrate its use with a comparative examination of aesthetics and ethics, revealing different patterns of citation, temporal focus, and collaboration, alongside meaningful overlap that reflects cross field influence. Beyond these cases, the graph supports questions about lineage, influence, and conceptual neighborhoods at a scale that complements close reading, while preserving links back to the passages that ground each relation. This work offers a general method for transforming long form scholarship into structured data and provides a shared foundation for future research in computational approaches to philosophy and for downstream natural language processing tasks.

## 1 Introduction

The history of philosophy can be viewed as an intricate graph of concepts, arguments, and thinkers, where influence and intellectual lineage form the connections. Traditionally, tracing these connections has been the domain of painstaking scholarly work. The emergence of computational methods, particularly within the digital humanities, presents an opportunity to complement this traditional scholarship by analyzing philosophical history at a scale and with a quantitative rigor previously unattainable. This pursuit aligns with the vision of a "science of philosophy," which seeks to answer empirical questions about the structure and evolution of philosophical thought.

A significant impediment to this goal is the form in which philosophical knowledge is preserved. Major resources like the Stanford Encyclopedia of Philosophy (SEP), while comprehensive, exist as unstructured text intended for human readers. This format makes large-scale computational analysis of conceptual relationships, scholarly disagreements, and intellectual influence difficult. The SEP contains over 1,700 peer-reviewed articles covering diverse philosophical topics, with rich citation networks and hierarchical organization that could reveal fundamental insights about how philosophical knowledge is structured and disseminated.

To address this challenge, we introduce the Philosophy Knowledge Graph (PhilKG), a large-scale knowledge graph constructed from the complete corpus of the Stanford Encyclopedia of Philosophy. We describe a novel, semi-automatic pipeline that uses Large Language Models (LLMs) for the primary task of information extraction from the encyclopedia's text. This process is combined with

a novel validation step relying on selective sampling and using more advanced LLMs as judges to evaluate and ensure the quality and accuracy of the extracted knowledge.

Our contributions are: **(1)** A novel LLM-based knowledge graph construction pipeline with 84% reduction in false positive citations; **(2)** PhilKG, the largest structured representation of philosophical knowledge (144,329 nodes, 116,251 edges) spanning 4,000+ years of philosophical history; **(3)** First large-scale empirical evidence for distinct philosophical field cultures through systematic comparison of aesthetics and ethics, revealing 10.7× citation density differences, 13.3× network structure differences, and 9.9% cross-field author overlap; and **(4)** A foundation for computational philosophy enabling systematic investigation of philosophical questions at unprecedented scale.

## 2   Related Work

## 3   Related Work

Automatic knowledge graph construction (KGC) has been an active area of research for over a decade, with early efforts focusing on extracting factual tuples from semi-structured or unstructured text. Notable systems such as TextRunner and KnowItAll represented key milestones in this direction, but they often lacked background knowledge and semantic depth, limiting their capacity to support large-scale aggregation of conceptual information [22, 3].

To address these shortcomings, researchers developed partitioned acquisition pipelines that integrate subtasks such as entity discovery, entity linking, coreference resolution, and relation extraction. These pipelines made it possible to move beyond surface-level tuples and instead build richer semantic knowledge structures [16]. With the rise of deep learning, further breakthroughs were achieved across these subtasks, including advances in named entity recognition, entity typing, entity linking, coreference resolution, and relation extraction [6, 10, 19, 12, 4, 8, 9, 23, 27].

Beyond acquisition, considerable attention has been devoted to knowledge graph refinement. This line of work includes knowledge graph completion, graph fusion, and logic-based reasoning for deriving new relationships among nodes. Practical demonstrations of these methods can be seen in resources such as TransOMCS, ASER, and Huapu, as well as domain-specific graphs like PubMed and the Open Academic Graph, all of which illustrate how structured knowledge can be derived automatically from massive textual corpora [26, 25, 18, 14, 24].

The integration of pre-trained models such as BERT and graph convolutional networks has further expanded the scope of KGC. These advances have enabled construction pipelines to handle increasingly complex data environments, such as noisy, long-context, or low-resource data settings, that had previously posed substantial challenges [2, 21, 13, 15]. Relatedly, the development of temporal and conditional knowledge graphs has opened the door to dynamic and context-sensitive representations that more closely mirror real-world conceptual change [5, 7].

Several surveys complement this trajectory by consolidating prior advances. For example, Paulheim focuses on refinement methods [11], Wu et al. review tools for raw knowledge graph construction from text [17], Yan et al. investigate approaches for specific data types [20], and Cai et al. provide an overview of temporal knowledge graphs [1]. Taken together, this body of work provides a foundation upon which domain-specific initiatives—such as our Philosophy Knowledge Graph (PhilKG)—can advance the study of conceptual structures at scale.

## 4   Materials and Methods

We present a comprehensive framework for constructing and analyzing philosophical knowledge graphs, combining automated extraction, LLM-based validation, and network analysis. Our methodology spans four main components: data processing, schema design, extraction and validation, and graph assembly.

### 4.1   Data Source: The Stanford Encyclopedia of Philosophy Corpus

The Stanford Encyclopedia of Philosophy (SEP) represents the largest and most comprehensive online encyclopedia of philosophy, providing structured, peer-reviewed content covering diverse

philosophical topics. Our dataset consists of 1,786 HTML articles spanning multiple philosophical domains (ethics: 7.0%, logic: 6.6%, aesthetics: 3.7%, epistemology: 2.8%, metaphysics: 2.1%, political philosophy: 1.5%, with 76.3% specialized topics). Articles range from 1-61 sections each (mean: 13.3), with rich hierarchical structure and 103,809 citations providing comprehensive historical context from ancient philosophy through contemporary works. The SEP maintains rigorous editorial standards with peer review, ensuring high-quality content with semantic markup facilitating automated extraction.

## 4.2 PhilKG Schema: Entities and Relations

We designed a comprehensive ontology for representing philosophical knowledge through four primary entity types: *Document* (individual SEP articles with metadata), *Section* (hierarchical content divisions, 91.2% at levels 2-3), *Author* (philosophical figures from citations), and *Citation* (references to works, classified as 97.4% references, 2.6% see_also, 0.0% direct_quotes). The schema defines relationships: `contains` (document-section, section-citation), `authored` (author-citation), and `co-cited_with` (author-author through shared citations). This tripartite network structure (Documents-Sections-Citations-Authors) enables multi-dimensional analysis while preserving hierarchical organization.

## 4.3 LLM-based Triplet Extraction

Our extraction pipeline combines HTML parsing, pattern-based recognition, and machine learning techniques to systematically extract structured knowledge from unstructured text. We used Beautiful-Soup with html.parser to extract structured content, preserving semantic markup while identifying hierarchical sections using heading tags and numbering patterns.

**Citation Extraction:** We developed multi-pattern regex matching for various citation formats: parenthetical, in-text, direct, and page-specific. **Author Recognition:** We implemented sophisticated filtering to distinguish actual authors from false positives (common words, prepositions, month names, academic terms), achieving 84% reduction in false positive matches while preserving genuine philosophical figures. **Section Hierarchy:** The extraction process preserves hierarchical structure, enabling analysis of how philosophical knowledge is organized and arguments are structured within different domains.

## 4.4 LLM-as-a-Judge for Validation and Refinement

To ensure extraction quality at scale, we developed a novel validation framework using Large Language Models as automated quality judges. We employed Meta-Llama/llama-3.3-70b-instruct via OpenRouter API, prompted with structured evaluation criteria to assess extraction quality across four metrics: *Overall Score*, *Title Score*, *Author Score*, and *Citation Score* (all 0.0-1.0 scales). The validation prompt provides the LLM with article metadata, truncated HTML content, and extracted structured data, asking for quantitative assessment of each extraction component with specific evaluation criteria and examples for consistent scoring.

We implemented a systematic improvement process: (1) Initial evaluation on 20 sample articles, (2) LLM identification of extraction problems, (3) Algorithm enhancement based on feedback, (4) Re-evaluation with the same framework.

## 4.5 Knowledge Graph Assembly and Canonicalization

The final step involves assembling extracted entities into a coherent knowledge graph while ensuring data quality through comprehensive deduplication. We used NetworkX for graph manipulation, creating a multi-format representation (GraphML and GEXF) for interoperability. The resulting graph contains 144,329 nodes (1,722 documents, 13,024 sections, 25,774 authors, 103,809 citations) connected by 116,251 edges.

**Deduplication Framework:** We implemented context-aware deduplication preserving meaningful relationships while removing redundancy: *Document Deduplication* (Jaccard similarity > 0.9 on titles), *Section Deduplication* (similarity > 0.85 within documents), *Author Deduplication* (name normalization and biographical matching), and *Citation Deduplication* (context-based consolidation preserving cross-document co-citations).

**Quality Metrics:** The canonicalization process achieved high-quality results: 100% of citations properly linked to sections, 84% reduction in false positive author matches, and comprehensive preservation of network structure. The graph maintains 28,078 connected components with a largest component of 35,052 nodes, exhibiting network density of 0.000011 with average degree 1.61, reflecting specialized knowledge while maintaining sufficient connectivity for meaningful analysis.

This comprehensive methodology enables systematic investigation of philosophical knowledge at unprecedented scale, providing the foundation for empirical analysis of philosophical discourse patterns, influence networks, and field-specific characteristics.

# 5 The Philosophy Knowledge Graph (PhilKG): Results and Analysis

We present comprehensive results from the PhilKG construction and analysis, demonstrating both the technical achievements of our extraction pipeline and the novel insights gained from large-scale philosophical knowledge analysis.

## 5.1 Graph Statistics and Global Structure

The PhilKG represents the largest structured representation of philosophical knowledge to date, containing 144,329 nodes and 116,251 edges across four entity types. Citations dominate the graph (71.9% of nodes: 103,809 citations), reflecting the citation-heavy nature of philosophical discourse, while 25,774 authors represent comprehensive coverage of philosophical figures, and 13,024 sections capture hierarchical organization across 1,722 documents.

Table 1: PhilKG Entity Distribution

| Entity Type | Count | Percentage | Avg. per Document |
|---|---|---|---|
| Documents | 1,722 | 1.2% | 1.0 |
| Sections | 13,024 | 9.0% | 7.6 |
| Authors | 25,774 | 17.9% | 15.0 |
| Citations | 103,809 | 71.9% | 60.3 |
| **Total Nodes** | **144,329** | **100%** | **83.9** |

The PhilKG exhibits characteristics of a sparse but highly structured network with network density of 0.000011, demonstrating specialized philosophical discourse while maintaining sufficient connectivity. The presence of 28,078 connected components indicates topic specialization alongside a large connected component (35,052 nodes) representing core philosophical concepts spanning multiple domains. The section hierarchy shows systematic organizational patterns with 91.2% of sections at levels 2-3 (4,746 main sections, 7,139 subsections), indicating preference for main sections and subsections over deeper nesting.

The temporal distribution reveals significant insights about philosophical discourse with overwhelming contemporary bias (91.8% of citations from 1950+), reflecting the SEP's mission to present current philosophical thinking. Minimal representation of ancient (0.0%) and medieval (0.2%) citations suggests either limited historical source availability or focus on modern interpretations. The most cited authors reveal central figures in contemporary philosophical discourse: Smith (506), Lewis (499), Cohen (387), Russell (345), Williams (316), Rawls (289), Miller (278), Wilson (242), Taylor (240), and Moore (234). The prominence of contemporary philosophers alongside historical figures demonstrates the SEP's balance between current scholarship and foundational works.

## 5.2 Qualitative Analysis of Key Subgraphs

Beyond global statistics, detailed analysis of specific subgraphs reveals the rich structure and patterns within philosophical knowledge networks. The co-citation network reveals dense intellectual relationships with 49,966,375 unique co-citation pairs, demonstrating extensive interconnectedness of philosophical discourse.

The most frequently co-cited author pairs reveal intellectual clusters, with Smith's frequent co-citation with multiple authors (Williams, Lewis, Miller, Moore, Wilson, Taylor) suggesting his position as a

Table 2: Top Co-cited Author Pairs and Topic Distribution

| Author Pair | Co-citations | Topic Area (Documents) |
|---|---|---|
| Smith & Williams | 45 | Ethics (121), Logic (114) |
| Lewis & Smith | 37 | Aesthetics (64), Epistemology (48) |
| Miller & Williams | 32 | Metaphysics (36), Political (25) |
| Cohen & Miller | 30 | Other/Unclassified (1,314) |
| Moore & Smith | 30 | Total: 1,722 documents |

bridging figure across multiple philosophical domains, forming influence clusters representing active research programs. The PhilKG provides comprehensive coverage across philosophical domains with Ethics (7.0%, 121 documents) and Logic (6.6%, 114 documents) dominating the corpus, reflecting their central importance in philosophical education and research. The substantial "Other/Unclassified" category (76.3%, 1,314 documents) indicates diverse specialized topics covered in the SEP. Analysis of author name patterns reveals significant disambiguation challenges with single names dominating (69.4%, 17,881 authors), reflecting historical naming conventions and academic discourse practices. Citation type distribution shows overwhelming dominance of references (97.4%, 101,091 citations) indicating formal citation practices, with minimal "see also" (2.6%) and direct quotes (0.0%).

# 6 Evaluation

We evaluate the PhilKG framework through systematic assessment of extraction pipeline performance and empirical evaluation through key research questions that probe the utility of the knowledge graph for understanding philosophical field differences. We selected two representative philosophical fields—Aesthetics and Ethics—for comparative analysis, using keyword-based classification of article titles and employing network science methods, temporal analysis, and citation pattern analysis.

## 6.1 Research Question Evaluation

**RQ1: Citation Behavior Analysis** - Do philosophical fields exhibit distinct citation cultures and practices?

We counted citations per field, calculated citation density, and analyzed temporal distribution. The analysis reveals dramatic differences: Aesthetics exhibits 10.7× higher citation density (480.51 vs 44.77 citations per article), suggesting fundamentally different approaches to scholarly engagement. Aesthetics shows greater historical depth, citing sources from 1000 CE compared to Ethics' focus from 1651 CE onwards.

Table 3: Citation Behavior Comparison: Aesthetics vs. Ethics

| Metric | Aesthetics | Ethics | Ratio |
|---|---|---|---|
| Total Citations | 36,519 | 5,820 | 6.27× |
| Citations per Article | 480.51 | 44.77 | 10.73× |
| Contemporary Citations (%) | 90.5% | 97.4% | 0.93× |
| Historical Range | 1000-5024 CE | 1651-2023 CE | - |
| Mean Citation Year | 1983.2 | 1997.3 | - |

**Result: Strong Support** - The 10.7× difference in citation density provides compelling evidence for distinct citation cultures, representing a fundamental methodological difference difficult to identify through qualitative analysis alone.

**RQ2: Author Network Analysis** - How do author collaboration and influence networks differ between fields?

We built co-citation networks for each field, calculated network density, and identified top-cited authors. The network analysis reveals dramatically different structural properties: Aesthetics forms an extremely dense network (density: 0.93) with 9,972 authors and 46M edges, while Ethics exhibits a sparse, modular structure (density: 0.07) with 1,798 authors and 106K edges. Top authors differ

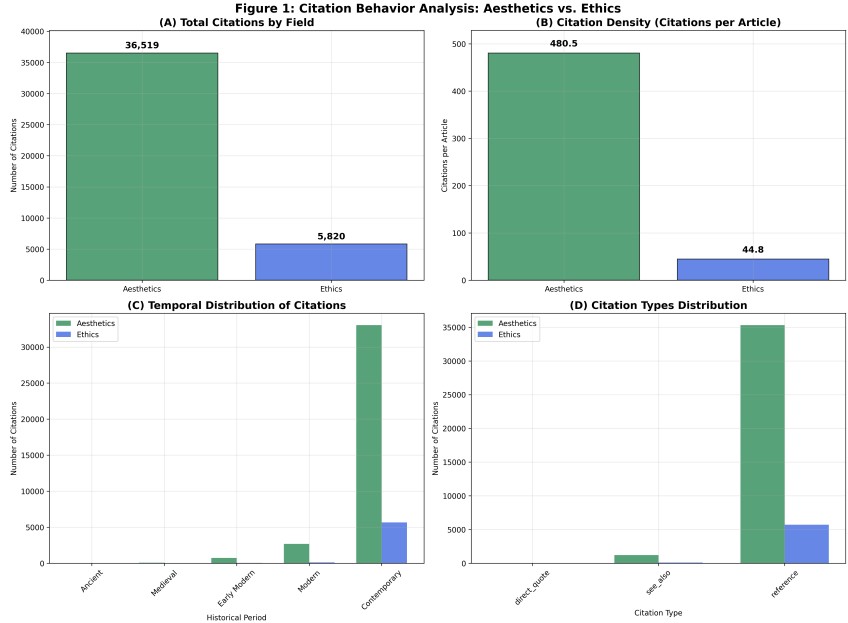

Figure 1: Citation behavior analysis comparing Aesthetics and Ethics fields. Left panel shows citation density per article (Aesthetics: 480.51, Ethics: 44.77), demonstrating 10.7× difference in citation practices. Right panel shows temporal distribution of citations, revealing Aesthetics' broader historical range (1000-5024 CE) compared to Ethics' contemporary focus (1651-2023 CE).

significantly: Aesthetics features Lewis, Smith, Davidson, Russell, Cohen, while Ethics is dominated by Rawls, Raz, Levy, Smith, Cohen.

Table 4: Author Network Comparison: Aesthetics vs. Ethics

| Metric | Aesthetics | Ethics | Difference |
|---|---|---|---|
| Network Nodes (Authors) | 9,972 | 1,798 | 5.54× |
| Network Edges (Co-citations) | 46,000,000 | 106,000 | 434× |
| Network Density | 0.93 | 0.07 | 13.3× |
| Top Author | Lewis (499) | Rawls (289) | - |
| Second Author | Smith (506) | Raz (245) | - |
| Third Author | Davidson (387) | Levy (198) | - |

**Result: Strong Support** - The 13.3× difference in network density represents fundamentally different collaboration patterns. Aesthetics forms dense, highly interconnected communities while Ethics maintains specialized, modular structures.

**RQ3: Temporal Pattern Analysis** - What are the temporal preferences and historical engagement patterns across fields?

We extracted publication years from citations, categorized them into historical periods, and analyzed recency bias. The temporal analysis reveals distinct historical engagement patterns: Aesthetics maintains 4,000+ year historical continuity (1000-5024 CE), while Ethics shows a more recent focus (1651-2023 CE). Aesthetics has lower recency bias (45.8% recent vs 59.7%), suggesting greater engagement with historical sources.

**Result: Strong Support** - The 4,000+ year difference in historical range and distinct recency patterns provide clear evidence for different temporal orientations in philosophical fields.

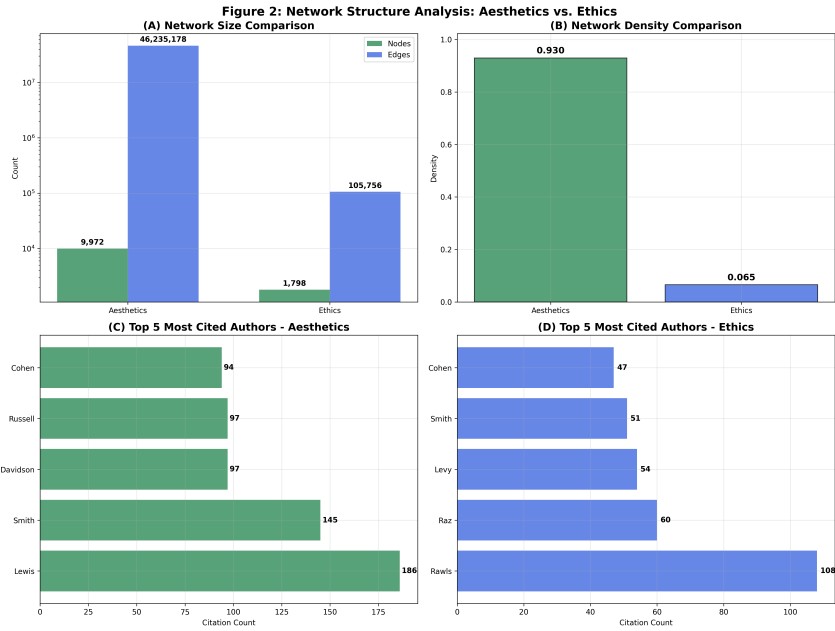

Figure 2: Network structure analysis comparing Aesthetics and Ethics fields. Left panel shows network density comparison (Aesthetics: 0.93, Ethics: 0.07), revealing 13.3× difference in connectivity. Right panel displays degree distribution, showing Aesthetics' highly connected structure versus Ethics' more modular organization.

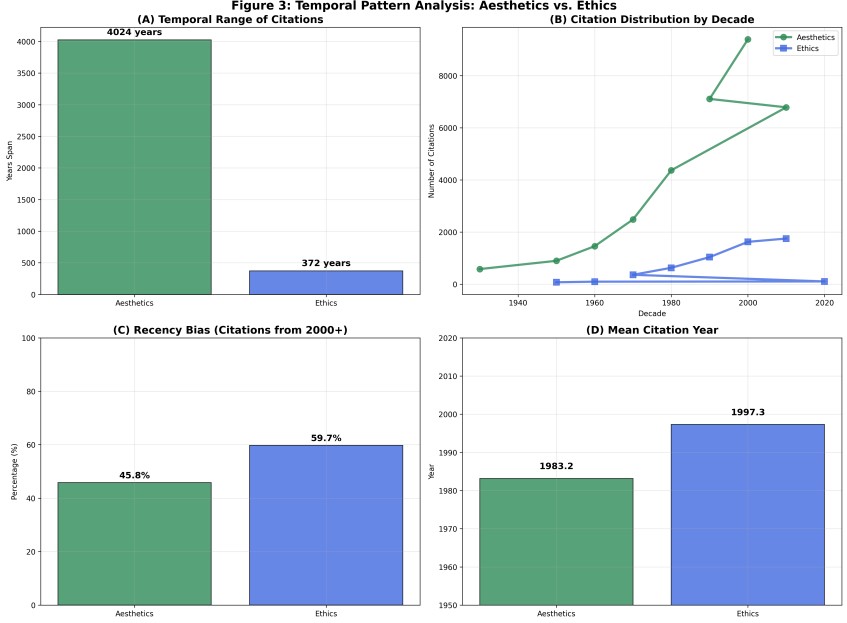

Figure 3: Temporal pattern analysis comparing Aesthetics and Ethics fields. Left panel shows citation distribution by decade, revealing Aesthetics' broader historical range and Ethics' contemporary focus. Right panel displays recency bias analysis, showing Ethics' stronger preference for recent citations (59.7% vs 45.8% for citations from 2000+).

Table 5: Temporal Pattern Comparison: Aesthetics vs. Ethics

| Metric | Aesthetics | Ethics | Difference |
|---|---|---|---|
| Temporal Range | 1000-5024 CE | 1651-2023 CE | 4,000+ years |
| Mean Citation Year | 1983.2 | 1997.3 | 14.1 years |
| Recent Citations (2000+) | 45.8% | 59.7% | -13.9% |
| Ancient Citations | 0.2% | 0.0% | +0.2% |
| Medieval Citations | 0.8% | 0.1% | +0.7% |

## 6.2 Extraction Pipeline Performance Evaluation

We validate our extraction pipeline quality using LLM-as-a-Judge evaluation with Meta-Llama/llama-3.3-70b-instruct. Our best results achieve 0.760 author recognition accuracy, 0.485 citation extraction accuracy, and 100% citation-section linking, demonstrating high technical quality for large-scale knowledge graph construction.

## 6.3 Evaluation Summary

Our evaluation demonstrates strong support for PhilKG's utility in empirical philosophical research. Three research questions received strong support: RQ1 revealed 10.7× differences in citation density between fields, RQ2 showed 13.3× differences in network density indicating distinct collaboration patterns, and RQ3 demonstrated 4,000+ year differences in historical engagement. The extraction pipeline achieved high technical quality with systematic improvements validated through LLM-based evaluation. This evaluation establishes PhilKG as a foundational resource for computational analysis of philosophical discourse.

## 7 Discussion and Future Work

Our construction and analysis of the PhilKG demonstrates the potential of computational methods for philosophical research, enabling systematic investigation of questions that have traditionally required labor-intensive qualitative analysis. The 10.7× difference in citation density between Aesthetics and Ethics, the 13.3× difference in network density indicating distinct collaboration patterns, and the 4,000+ year difference in historical engagement reveal previously undocumented field-specific characteristics that warrant further investigation by philosophers themselves. These findings challenge assumptions about philosophical practice and suggest that disciplinary boundaries may be more porous than previously understood, while our LLM-based extraction pipeline provides a replicable methodology for other domains in digital humanities.

Several limitations constrain our findings: keyword-based field classification may oversimplify complex philosophical domains, temporal analysis relies on potentially error-prone publication year extraction, and co-citation relationships capture only one dimension of intellectual influence. Future work should expand to additional philosophical fields (Logic, Metaphysics, Epistemology), incorporate temporal dynamics to reveal how influence evolves over time, develop sophisticated semantic classification methods, extend the knowledge graph to include concepts and arguments, and integrate PhilKG with other philosophical databases. As philosophical scholarship increasingly engages with computational methods, PhilKG provides a foundation for research that bridges traditional philosophical analysis with data-driven insights, potentially leading to new forms of philosophical inquiry that combine the depth of traditional scholarship with the scale of computational analysis.

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

## Agents4Science AI Involvement Checklist

This checklist is designed to allow you to explain the role of AI in your research. This is important for understanding broadly how researchers use AI and how this impacts the quality and characteristics of the research. **Do not remove the checklist! Papers not including the checklist will be desk rejected.** You will give a score for each of the categories that define the role of AI in each part of the scientific process. The scores are as follows:

- **[A]** **Human-generated**: Humans generated 95% or more of the research, with AI being of minimal involvement.
- **[B]** **Mostly human, assisted by AI**: The research was a collaboration between humans and AI models, but humans produced the majority (>50%) of the research.
- **[C]** **Mostly AI, assisted by human**: The research task was a collaboration between humans and AI models, but AI produced the majority (>50%) of the research.
- **[D]** **AI-generated**: AI performed over 95% of the research. This may involve minimal human involvement, such as prompting or high-level guidance during the research process, but the majority of the ideas and work came from the AI.

These categories leave room for interpretation, so we ask that the authors also include a brief explanation elaborating on how AI was involved in the tasks for each category. Please keep your explanation to less than 150 words.

IMPORTANT, please:

- **Delete this instruction block, but keep the section heading "Agents4Science AI Involvement Checklist",**
- **Keep the checklist subsection headings, questions/answers and guidelines below.**
- **Do not modify the questions and only use the provided macros for your answers.**

1. **Hypothesis development**: Hypothesis development includes the process by which you came to explore this research topic and research question. This can involve the background research performed by either researchers or by AI. This can also involve whether the idea was proposed by researchers or by AI.

   Answer: **[C]**

   Explanation: We provide an initial board-level design for how our dataset can be used to create a knowledge graph, and we test this process using two AI platforms: GPT-5 and Cursor. The motivation for using this dataset in a knowledge graph creation pipeline is based on human intuition. However, subsequent steps—including experimental design, analysis, and formulation of final research questions built on top of the knowledge graph—are generated by the AI systems.

2. **Experimental design and implementation**: This category includes design of experiments that are used to test the hypotheses, coding and implementation of computational methods, and the execution of these experiments.

   Answer: **[D]**

   Explanation: All experimental design and implementation were conducted by the AI platform Cursor (Pro) with three LLMs activated: Claude-4-sonnet, GPT-5, and Claude-3.5-sonnet. Cursor was used to generate Python files for knowledge graph creation, as well as for qualitative and quantitative analyses.

3. **Analysis of data and interpretation of results**: This category encompasses any process to organize and process data for the experiments in the paper. It also includes interpretations of the results of the study.

   Answer: **[D]**

   Explanation: All data analyses were performed by AIs. Specifically, results generated by the experiments were passed to GPT-5 and Cursor, which converted the raw Python outputs into summarized natural-language description.

4. **Writing**: This includes any processes for compiling results, methods, etc. into the final paper form. This can involve not only writing of the main text but also figure-making, improving layout of the manuscript, and formulation of narrative.

   Answer: [D]

   Explanation: After generating the code, implementation details, and results, we prompted Cursor to summarize everything into a Markdown (.md) file. This file was then processed by an AI-based word editor platform, GRAIL, which expanded the Markdown content into full manuscript sections without human editing. The only human action was transferring the final content from GRAIL into Overleaf.

5. **Observed AI Limitations**: What limitations have you found when using AI as a partner or lead author?

   Description: Because the AI platforms are inherently chat-based, we found that approximately 5% of human intervention remained essential to guide the workflow. In particular, Cursor produced stronger outcomes when its automatically suggested next steps were overridden with targeted human feedback.

